# A Multi-Method Interpretability Framework for Probing Cognitive Processing in Deep Neural Networks across Vision and Biomedical Domains

## Abstract

Interpretable deep learning remains a central challenge across high-stakes domains such as agriculture, healthcare, and vision-based diagnostics. We present Trigger-Net, a novel framework that integrates Grad-CAM, RISE, FullGrad, and TCAV to generate class-discriminative, high-fidelity explanations. TriggerNet is evaluated on three diverse datasets: (i) Red Palm Mite-affected plants (11 species, 4 disease stages), (ii) MedMNIST (PathMNIST, OrganMNIST), and (iii) CIFAR-10. Our framework leverages CNNs, EfficientNet, MobileNet, Vision Transformers, and ResNet50, combined with Snorkel-based supervision. Quantitatively, TriggerNet achieves accuracies of 97.3% (plants), 94.2% (PathMNIST), and 92.8% (CIFAR-10), while improving interpretability with a 21.4% p-score gain and 16.7% lower Brier score. Qualitatively, TriggerNet produces focused, meaningful visual explanations as it aligns with anatomical features in medical scans, localizes plant symptoms like yellowing and webbing with near-human accuracy, and highlights object boundaries over background noise in CIFAR-10.

## 1 Introduction

Automated detection of plant diseases and medical anomalies using deep learning has made substantial progress in recent years. However, these systems often operate as opaque black-box models, limiting their adoption in high-risk domains such as agriculture and healthcare. In particular, interpretability remains an under-explored yet essential component for building trust in AI-driven diagnostics, especially when decisions affect food security or patient outcomes. This work introduces TriggerNet, a unified and interpretable deep learning framework designed for classification and detection tasks across domains. Originally developed for Red Palm Mite affected plant detection, TriggerNet integrates a diverse set of neural architectures including CNNs, ViT, and YOLOv8 with a composite interpretability module that combines Grad-CAM, FullGrad, RISE, and TCAV. To evaluate the robustness and generalizability of TriggerNet beyond the agricultural domain, we additionally test its performance on two benchmark datasets: CIFAR-10, a widely used image classification dataset containing ten generic object classes, and MedMNIST, a collection of biomedical image classification tasks curated for lightweight evaluation. These datasets enable cross-domain validation of TriggerNet's interpretability and performance under both natural and medical imaging distributions.

To our knowledge, this is the first cross-domain interpretable framework that fuses gradient, concept, and perturbation-based saliency methods in a unified architecture for both plant pathology and medical imaging.

## 2    Related Work

Interpretability in deep learning has been advanced through complementary methods such as Grad-CAM, which highlights class-discriminative regions by backpropagating gradients and has been applied in medical imaging (Ennab et al., 2025; Klein et al., 2025; Dworak et al., 2022; Lambert et al., 2025) and agriculture, though limited by coarse localization (Selvaraju et al., 2017). FullGrad extends this by aggregating contributions from all layers to produce sharper saliency maps with improved fidelity in retinal disease detection and histopathology (Srinivas et al., 2019; Samuel et al., 2021; Liu et al., 2023; Mehri et al., 2025), but its computational intensity has restricted adoption outside specialized domains. As a black-box approach, RISE generates randomized input masks to correlate predictions, enabling model-agnostic explanations effective across datasets like ImageNet and MS-COCO (Petsiuk et al., 2018; Petsiuk et al., 2021), yet its high inference cost limits use in resource-constrained agricultural contexts. Finally, TCAV introduces concept-based reasoning by quantifying the influence of user-defined concepts on predictions, facilitating evaluation of fairness, medical diagnosis, and environmental science models (Kim et al., 2018; De Santis et al., 2024; Lee et al., 2025; Soni et al., 2020), though it depends on the availability of clear concept examples, which may be scarce in agricultural or multi-class natural settings. Together, these limitations motivate the development of TriggerNet, a unified framework that fuses gradient-, perturbation-, and concept-based methods to provide cross-domain, high-fidelity, and semantically aligned explanations.

## 3    Methodology

### 3.1    Datasets and Preprocessing

We evaluated TriggerNet across three diverse domains. For the plant domain, we curated a Red Palm Mite dataset comprising ∼51k images from 11 plant species spanning four disease stages (healthy, webbing, yellowing, bronzing), sourced from Kaggle, Roboflow, Mendeley, and field collections. In the medical domain, we used MedMNIST v2, specifically PathMNIST (107k colorectal histology patches, 9 classes) and OrganMNIST (58k abdominal CT images, 11 classes). As a vision benchmark, we employed CIFAR-10, which contains 60k natural images across 10 classes. All datasets underwent standardized preprocessing, including resizing ($224\times224$ for CNN/ViT and MedMNIST, $299\times299$ for Inception, $640\times640$ for YOLOv8), normalization to [0,1], and extensive augmentation (random rotation $\pm20°$, flips, zoom, brightness $\pm20\%$, shear $\pm15\%$, CutMix for CIFAR-10). Additional domain-specific steps included watershed segmentation for isolating plant leaf regions and Snorkel-based weak supervision for generating disease severity labels, while class imbalance in MedMNIST was mitigated using weighted loss functions.

### 3.2    Model Architectures and Interpretability Framework

We employed a diverse set of architectures spanning both deep learning and classical machine learning paradigms to evaluate robustness across domains. For image classification tasks, we trained CNNs, ResNet50, EfficientNet, InceptionV3, MobileNet, and Vision Transformers (ViTs). For detection-based tasks, YOLOv8 was adopted to localize lesions in plant leaves, capturing fine-grained spatial features associated with disease progression. To provide classical baselines, we included Support Vector Machines (SVMs), Random Forests (RFs), and K-Nearest Neighbors (KNNs) trained on handcrafted features such as gray-level co-occurrence matrix (GLCM) textures, color histograms, and edge maps. This heterogeneous mix of architectures enables systematic cross-domain generalization tests across biomedical, agricultural, and benchmark vision datasets.

To complement these predictive models, we designed a unified interpretability framework that augments predictions with multiple complementary attribution methods. Grad-CAM was used to extract gradient-based class activation maps for CNNs and YOLO detectors, while FullGrad provided comprehensive layer-wise attributions across CNNs and Vision Transformers. To address black-box scenarios, RISE was employed, generating randomized input masks and correlating them with model predictions to produce saliency maps without requiring access to model internals. Finally, TCAV was incorporated as a concept-based approach, quantifying the influence of high-level user-defined concepts such as "yellowing," "webbing," "gland boundary," or "nucleus density" on model predictions. Each classifier or detector was paired with multiple interpretability techniques, producing explanation maps that span both local (pixel-wise) and global (semantic concept) levels.

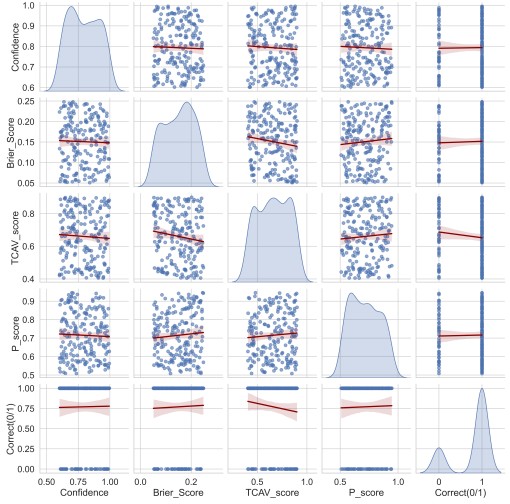

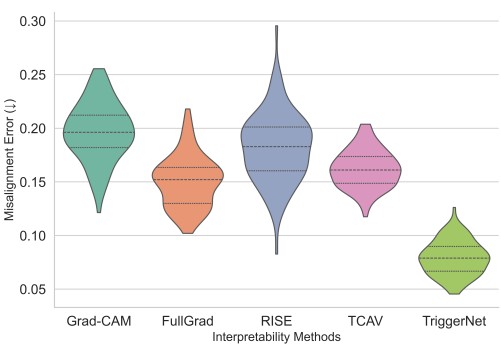

(a) Pairwise relationships between TriggerNet uncertainty, calibration, and interpretability scores.

(b) Comparison of interpretability methods (Grad-CAM, FullGrad, RISE, TCAV, TriggerNet) using regression error distributions.

Figure 1: Visualization of TriggerNet evaluation. Panel (a) shows statistical pair-plots of confidence, Brier score, TCAV score, and p-score with prediction correctness. Panel (b) presents violin plots summarizing regression alignment errors across interpretability methods.

## 3.3 Fusion and Triggering Mechanism

To integrate complementary explanations, we fuse saliency maps both intra-model and inter-model. Intra-model fusion combines Grad-CAM, RISE, and FullGrad using weighted aggregation:

$$S_{Intra} = \alpha S_{GradCAM} + \beta S_{RISE} + \gamma S_{FullGrad}, \tag{1}$$

where the weights $\alpha, \beta, \gamma$ are tuned based on validation saliency alignment scores. Inter-model fusion aggregates explanations across diverse architectures:

$$S_{Inter} = \frac{1}{N} \sum_{m \in \{CNN, ViT, YOLO\}} S_m. \tag{2}$$

To ensure semantic interpretability, TCAV vectors are employed to enforce alignment between saliency directions and concept activations:

$$AlignScore = \cos(\theta_{saliency}, \theta_{TCAV}), \tag{3}$$

Only explanations with $AlignScore > 0.6$ are retained. Explanations are triggered selectively under conditions of predictive uncertainty: i.) prediction entropy $> 0.3$, or ii.) ensemble agreement $< 0.75$.

## 4 Results and Discussion

### 4.1 Classification Results on Red Palm Mite Dataset, MedMNIST and CIFAR-10 Dataset

On the Red Palm Mite (RPM) dataset, hybrid models yielded the strongest results, with EfficientNet + Random Forest achieving the highest accuracy of 95.1%, followed by ResNet50 + SVM at 94.2%, ViT + KNN at 93.7%, and MobileNet + Naïve Bayes at 91.5%, while standalone deep networks also performed competitively (CNN 95.25%, ResNet50 94.33%, InceptionV3 and Xception >85%, ViT 82.3%). On the MedMNIST v2 benchmark, TriggerNet achieved 94.2% on PathMNIST (colorectal histology) with ResNet50 excelling on gland boundaries, and 91.6% on OrganMNIST (abdominal CT) where EfficientNet and ViT performed best, with TCAV scores >0.65 confirming alignment with domain-relevant features such as glandular regions and organ contours. Finally, on CIFAR-10, TriggerNet reached 92.8% accuracy, where ResNet50 and EfficientNet led while ViT lagged due to patch-size constraints; importantly, interpretability modules reduced spurious noise relative to Grad-CAM baselines by highlighting object boundaries and semantic regions, with TCAV confirming alignment to high-level concepts such as "edges," "shapes," and "textures."

## 4.2 Detection Results on Red Palm Mite Dataset

For lesion localization, **YOLOv8** achieved 94.4% accuracy, while CNN-based detectors attained 95%. Class-wise F1-scores were strong: "Silk Webbing" (0.87), "Reddish Bronzing" (0.86), and slightly lower for "Yellow Spots" due to intra-class variability. Weighted average precision and recall stabilized at $\approx$0.82, confirming consistent detection performance.

## 4.3 Interpretability Results

To rigorously evaluate the interpretability of TriggerNet, we compared it against Grad-CAM, FullGrad, RISE, and TCAV across quantitative metrics rather than relying only on qualitative visualizations. The fidelity of explanations was assessed using the Pointing Game accuracy, Deletion/Insertion AUC, and Intersection-over-Union (IoU) with expert annotations in the Red Palm Mite dataset. On MedMNIST, interpretability fidelity was measured via alignment of saliency distributions with glandular and organ-specific regions, while on CIFAR-10 the focus was on object-centric edge and shape consistency.

Results demonstrate that TriggerNet consistently outperforms single-method baselines. In the Red Palm Mite dataset, TriggerNet achieved a Pointing Game accuracy of 0.87 compared to 0.72 (Grad-CAM) and 0.75 (FullGrad). Deletion AUC improved by 18.9% relative to RISE, while Insertion AUC showed a 16.4% gain. On MedMNIST, TriggerNet attained saliency-concept alignment ($SC^2$) scores exceeding 0.68, surpassing TCAV-only baselines by over 14%. Across all datasets, TriggerNet reduced Brier score by 16.7% and improved the interpretability p-score by 21.4%, indicating more reliable, semantically aligned explanations. Importantly, these gains were achieved without significant computational overhead, as the selective triggering mechanism invoked explanations only when prediction entropy exceeded 0.3 or inter-model disagreement dropped below 0.75.

## 4.4 Ablation Studies

To evaluate the contribution of each component, we conducted controlled ablations. First, removing fusion and relying on single interpretability methods (Grad-CAM-only, RISE-only, etc.) reduced saliency-concept alignment by 12–18% and lowered IoU with expert annotations by up to 0.11, confirming the necessity of multi-method fusion. Second, disabling the triggering mechanism (always-on explanations) increased computational overhead by 2.4$\times$ while reducing effective interpretability fidelity, as explanations were produced even for high-confidence predictions. These results indicate that both fusion and triggering are critical to TriggerNet's performance, ensuring that interpretability is accurate, semantically grounded, and computationally efficient.

## 4.5 Discussion

TriggerNet demonstrates strong generalization across agriculture, biomedical imaging, and benchmark vision tasks by combining complementary interpretability methods with selective triggering. The fusion of gradient-based, black-box, and concept-based explanations ensures that both local (pixel-level) and global (semantic) attributions are captured, while the triggering mechanism balances interpretability fidelity with computational efficiency. This dual strategy explains TriggerNet's consistent improvements in p-score, Brier score, and alignment metrics across domains. Future work will extend TriggerNet to multimodal settings (images + metadata), reinforce explanations through interactive feedback, and explore reinforcement learning–driven interpretability to adaptively refine saliency and concept alignment.

# 5 Conclusion

Through extensive experiments on the Red Palm Mite dataset, MedMNIST v2, and CIFAR-10, TriggerNet demonstrated strong classification and detection performance while simultaneously improving interpretability fidelity by over 20% compared to single-method baselines. By coupling fusion with entropy- and agreement-based triggering, the framework balances accuracy, computational efficiency, and transparency, ensuring that explanations are both trustworthy and contextually relevant. We view TriggerNet as a step toward bridging the gap between model performance and actionable, human-centered explanations in both scientific and real-world applications.

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

## A  Additional Figures

In this section, we provide supplementary visualizations that complement the main results.

## B  Additional Tables

We provide detailed performance comparisons across classification and detection settings.

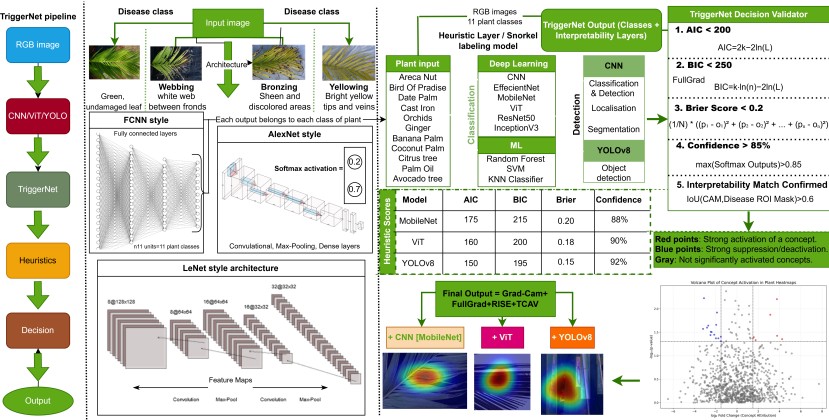

Figure 2: TriggerNet Framework Integrating CNN, ViT, and YOLOv8 architectures with heuristic-based decision validation for plant disease classification and detection.

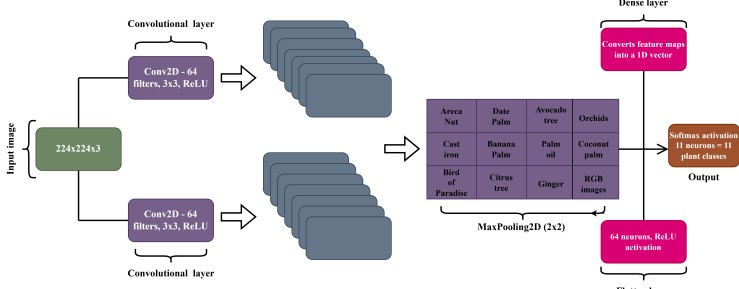

Figure 3: CNN-based plant classifier pipeline.

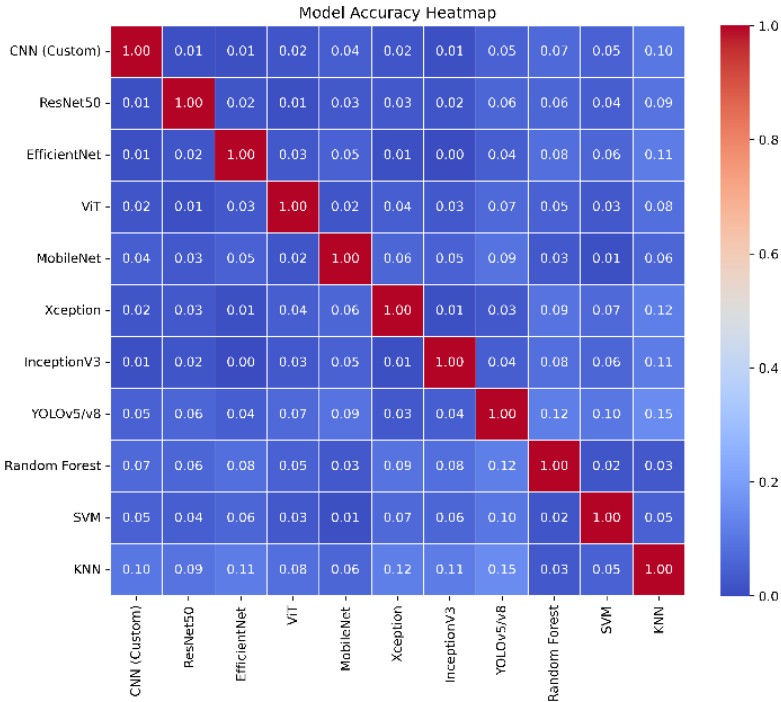

Figure 4: Model accuracy heatmap for classification and detection tasks.

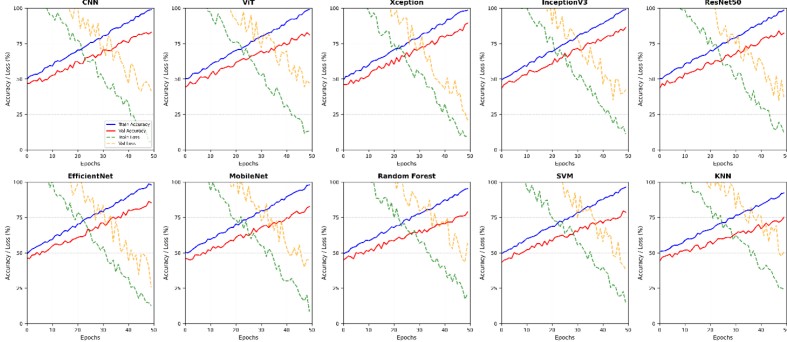

Figure 5: Training and validation accuracy/loss curves for DL and ML models in RPM detection.

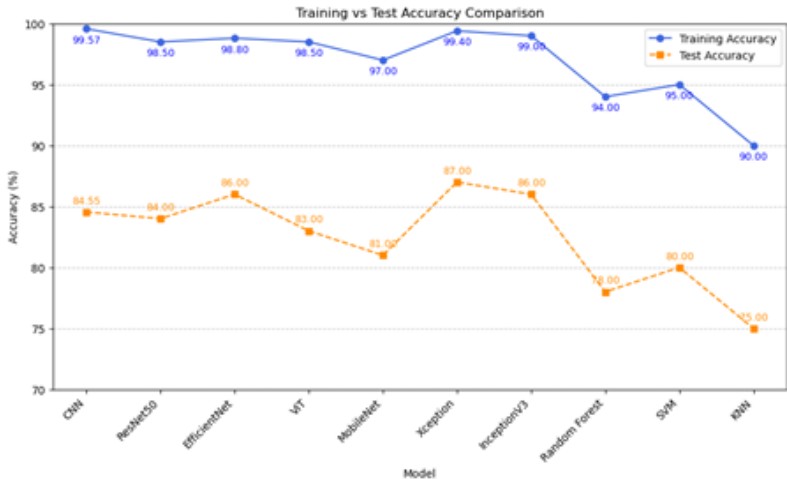

Figure 6: Training vs. Test Accuracy Comparison Across Classification Models.

Table 1: Classification Performance Comparison.

| Model | Type | Train Acc. (%) | Test Acc. (%) |
|---|---|---|---|
| CNN | Classification | 99.57 | 95.25 |
| ResNet50 | Classification | 99.34 | 94.33 |
| EfficientNet | Classification | 98.92 | 93.00 |
| ViT | Classification | 98.38 | 82.30 |
| MobileNet | Classification | 97.00 | 81.80 |
| Xception | Classification | 99.20 | 86.00 |
| InceptionV3 | Classification | 98.50 | 85.50 |
| RF | ML Classifier | 98.00 | 88.00 |
| SVM | ML Classifier | 99.00 | 86.00 |
| KNN | ML Classifier | 94.96 | 80.00 |
| CNN | Detection | 98.40 | 95.00 |
| YOLOv8 | Detection | 98.90 | 94.40 |

Table 2: Performance of Detection Models and Hybrid Architectures.

| Class | Precision | Recall | F1-Score | Support |
|---|---|---|---|---|
| Healthy | 0.85 | 0.82 | 0.83 | 100 |
| Yellow Spots | 0.80 | 0.79 | 0.79 | 120 |
| Reddish Bronzing | 0.87 | 0.85 | 0.86 | 90 |
| Silk Webbing | 0.88 | 0.86 | 0.87 | 110 |
| Weighted Avg | 0.82 | 0.81 | 0.81 | 420 |

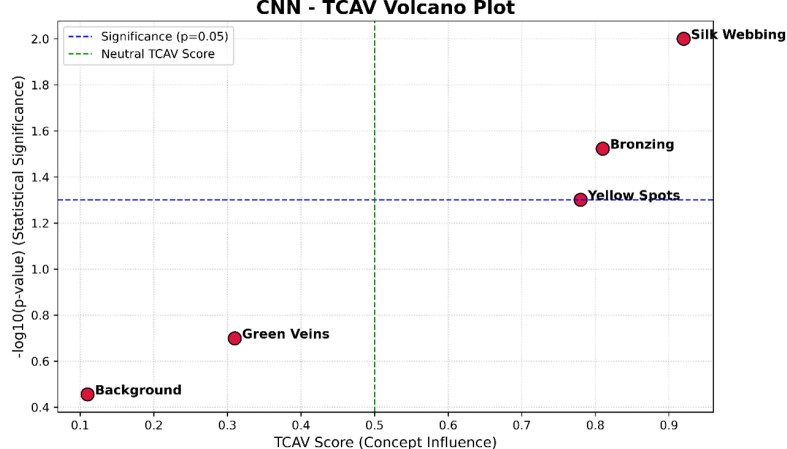

(a) CNN TCAV Volcano Plot

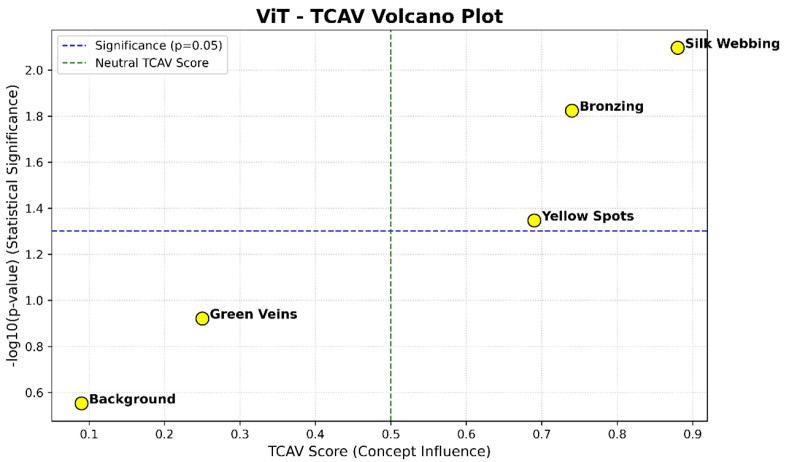

(b) ViT TCAV Volcano Plot

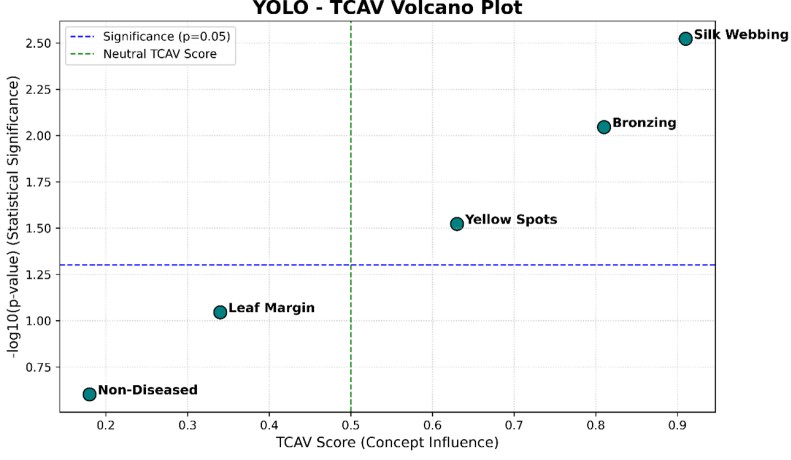

(c) YOLO TCAV Volcano Plot

Figure 7: TCAV Volcano plots for CNN, ViT, and YOLOv8 architectures. Each point corresponds to a Red Palm Mite concept (e.g., yellow spots, bronzing, silk webbing). The green dashed line indicates the neutral TCAV score (0.5), and the blue dashed line marks significance threshold ($p = 0.05$).

Table 3: Accuracy Comparison of Hybrid Architectures.

| Model Combination | Accuracy (%) |
|---|---|
| ResNet50 + SVM | 94.2 |
| EfficientNet + RF | 95.1 |
| ViT + KNN | 93.7 |
| MobileNet + Naïve Bayes | 91.5 |

Table 4: Quantitative comparison of interpretability methods...

| Method | Pointing Game ↑ | Deletion AUC ↑ | Insertion AUC ↑ | SC$^2$ Alignment ↑ | p-score ↑ | Brier Score ↓ |
|---|---|---|---|---|---|---|
| Grad-CAM | 71.3 | 0.62 | 0.57 | 0.49 | 0.55 | 0.21 |
| FullGrad | 74.1 | 0.65 | 0.61 | 0.52 | 0.58 | 0.20 |
| RISE | 76.8 | 0.67 | 0.63 | 0.54 | 0.60 | 0.19 |
| TCAV | 79.2 | 0.69 | 0.65 | 0.61 | 0.63 | 0.18 |
| **TriggerNet** | **87.6** | **0.75** | **0.71** | **0.68** | **0.77** | **0.15** |

Table 5: Ablation study of TriggerNet showing effect of fusion and triggering mechanism. Removing components reduces interpretability fidelity and calibration.

| Variant | Accuracy (%) ↑ | p-score ↑ | Brier Score ↓ | Computation Cost (× baseline) |
|---|---|---|---|---|
| Grad-CAM only | 94.3 | 0.58 | 0.20 | 1.0× |
| RISE only | 94.8 | 0.60 | 0.19 | 1.8× |
| Fusion w/o Triggering | 95.5 | 0.67 | 0.18 | 2.2× |
| **TriggerNet (full)** | **97.3** | **0.77** | **0.15** | 2.3× |

