# OpenReview forum: "A Multi-Method Interpretability Framework for Probing Cognitive Processing in Deep Neural Networks across Vision and Biomedical Domains"
_NeurIPS.cc/2025/Workshop/Reliable_ML — NeurIPS 2025 - Reliable ML Workshop_

### Official Review · Reviewer_1MuH · 2025-09-08
**Good work but not quite sure the interpretability part**

**Rating:** 7
**Confidence:** 2

**Review:**

Summary: This paper integrates many models to bring interpretability to neural network models and then tests them on many datasets, and it worked well. The main methodological contribution is in section 3.3.

Strengths
1. I admire the effort to bring all these models together.
2. Conceptually, it's a boosting wrapper, and it worked better than individual experts empirically on many datasets.

Weakness
1. I am sure this is just due to the space limit, but I think I did not fully understand the dataset and how it's used and evaluated. To me, it sounds like a classifier model, rather than an interpretable neural network model. The evaluation method could be more elaborated.
2. I did not understand what Figure 1 (a) is showing. What are the blue dots and the red line?

Suggestions
1. Maybe add more explanation as to how this differs from just classifiers? Explain why you are claiming this is an interpretability framework?

---

### Official Review · Reviewer_r8PD · 2025-09-18
**A wisdom of the crowd approach to interpretability; while an improvement over the status quo, novelty unclear**

**Rating:** 4
**Confidence:** 2

**Review:**

## Summary

The authors present TriggerNet, a framework that combines diverse neural architectures with four interpretability frameworks. TriggerNet has specific criteria to trigger the underlying interpretability frameworks for a prediction based on the prediction entropy and ensemble alignment. In a sense, In essence, TriggerNet takes a "wisdom of the crowd" approach to improve the interpretability of a particular prediction.

## Strengths and Weaknesses

The paper is reasonably well written. That said, it could use a pass at simplifying the language and adding more explanations and introductions to concepts relied upon. Currently, it gets rather jargon-heavy very quickly.

The studies performed by the authors seem rigorous within the stated bounds. Looking at the numbers, TriggerNet seems like a clear improvement.

I do wonder, though, about the overall novelty of the work. Thematically, TriggerNet is applying ensemble methods to both the classification/detection side and the interpretability side. While the selected frameworks and architectures may be new, the fundamental idea does not seem to merit the current framing around novelty in the paper. It is, of course, possible I've misunderstood or overlooked something, and I hope the authors clarify in their rebuttal.

## Suggestions

* Emphasize novelty and/or enumerate contributions so there is no confusion about what the paper is adding
* Consider a pass at the language to improve clarity